# Lessons We Have Learned Regarding Seroprevalence in High and Low SARS-CoV-2 Contexts in Greece before the Omicron Pandemic Wave

**DOI:** 10.3390/ijerph19106110

**Published:** 2022-05-17

**Authors:** Ourania S. Kotsiou, George D. Vavougios, Dimitrios Papagiannis, Elena Matsiatsiou, Dimitra Avgeri, Evangelos C. Fradelos, Dimitra I. Siachpazidou, Garifallia Perlepe, Angeliki Miziou, Athanasios Kyritsis, Eudoxia Gogou, Serafim Kalampokas, Georgios Kalantzis, Vaios S. Kotsios, Konstantinos I. Gourgoulianis

**Affiliations:** 1Faculty of Nursing, School of Health Sciences, University of Thessaly, GAIOPOLIS, 41110 Larissa, Greece; elenamatsiatsiou123@gmail.com (E.M.); avgeridimitra@hotmail.com (D.A.); efradelos@uth.gr (E.C.F.); 2Department of Respiratory Medicine, Faculty of Medicine, School of Health Sciences, University of Thessaly, BIOPOLIS, 41110 Larissa, Greece; dantevavougios@hotmail.com (G.D.V.); sidimi@windowslive.com (D.I.S.); perlepef19@gmail.com (G.P.); kellymiz95@yahoo.com (A.M.); thanoskyrit@hotmail.com (A.K.); egogou@uth.gr (E.G.); george.kalantzis4@gmail.com (G.K.); kgourg@med.uth.gr (K.I.G.); 3Public Health & Vaccines Lab, Department of Nursing, School of Health Sciences, University of Thessaly, GAIOPOLIS, 41110 Larissa, Greece; dpapajon@gmail.com; 4School of Health Sciences, University of Thessaly, GAIOPOLIS, 41110 Larissa, Greece; serkal@gmail.com; 5Metsovion Interdisciplinary Research Center, National Technical University of Athens, 44200 Athens, Greece; vaioskotsios@gmail.com

**Keywords:** antibody testing, Greece, semi-closed community, seroprevalence

## Abstract

Background: Antibody seroprevalence in rural communities remains poorly investigated. We compared the SARS-CoV-2 seroprevalence in two Greek communities in June and July 2021 after the end of the Delta-driven pandemic wave that started in November 2020. One community was affected worse than the other. Methods: The SARS-CoV-2 IgG II Quant method (Architect, Abbott, IL, USA) was used for antibody testing. Results: We found a high rate of SARS-CoV-2 seropositivity in both communities, approaching 77.5%. In the area with a higher burden of COVID-19, Malesina, seropositivity was achieved with vaccine-acquired and naturally acquired immunity, whereas in the low-burden context of Domokos, the high rates of seropositivity were achieved mainly with vaccination. Previously infected individuals were less likely to be vaccinated than previously uninfected adults. The antibody titers were significantly higher in previously infected, vaccinated participants than in unvaccinated ones. In total, 4% and 10% of the unvaccinated population were diagnosed seropositive for the first time while not knowing about the previous infection. Age and gender did not impact antibody titers in high- or low-burden contexts. Conclusions: Before the Omicron pandemic wave, herd immunity was reached in different contexts in Greece. Higher antibody titers were measured in infected vaccinated individuals than in infected unvaccinated ones.

## 1. Introduction

SARS-CoV-2 transmission dynamics and antibody seroprevalence in rural communities remain poorly investigated; these parameters could provide insight into the onset and development of subsequent pandemic waves and data to guide vaccination strategies. SARS-CoV-2 seroprevalence was low in the first year of the pandemic in Greece, where decisive public health measures were taken, including border restrictions, educational facility closures, and earlier adoption of mask-wearing [1]. Greece implemented a very restrictive policy immediately after the initiation of the pandemic [1].

In our previous studies, we showed that SARS-CoV-2 seropositivity approached 45% in high-burden contexts of COVID-19 in Greece after the easing of measures during the second pandemic wave in November 2020, which was exclusively attributed to humoral immunity [2]. Notably, when the vaccination strategy was included nine months after this pandemic wave, SARS-CoV-2 seropositivity was increased to 76% [3]. Vaccination alone generated an immune response in almost half of the population in this community. We found higher antibody titers in the case of vaccination in previously infected subjects than in unvaccinated ones [3]. In this direction, it has been shown that the antibody response to the first vaccine dose was almost twofold higher in individuals who were seropositive before vaccination compared to those who were seronegative, suggesting that prior infection primes the immune response to the first dose of mRNA-based vaccine [4].

Additional serosurveys are needed to monitor the local pandemic dynamic and estimate how far from herd immunity countries are that will guide the vaccination programs in 2022. Accordingly, in this study, we aimed to estimate seroprevalence in two large Greek communities in June and July 2021, respectively, after the end of the Delta-driven pandemic wave that started in November 2020 and continued up to January 2021 and before the ongoing Omicron pandemic wave that started in November 2021.

## 2. Materials and Methods

Two serosurveillance programs were conducted in the municipality of Malesina and Domokos from 14 June 2021 to 16 June 2021 and on 9 July 2021, respectively, to investigate the antibody responses to SARS-CoV-2 infection one month after the formidable second pandemic wave (lasting from November to May 2020) and a few months before the last Omicron pandemic wave initiated in November 2021 in Greece.

Malesina is a town and a former municipality in Phthiotis, Greece, with 4526 citizens [5], which was hit badly by the second pandemic wave [6]. Domokos is a town in Phthiotis, Greece, with 1531 citizens, which was not hit badly by the second pandemic wave [5]. All the residents of Malesina and Domokos were invited to participate in the program by the local authority and had been notified of the time and place thereof. Participants were recruited by announcing the research in the media while local officials organized a one-month recruitment campaign. Exclusion criteria were not considered as we aimed for a population-centered, epidemiological approach, i.e., an attempt to capture an accurate reflection of the population under study, rather than a specific subset. The participants were analyzed to evaluate seroprevalence due to SARS-CoV-2 infection and/or vaccination.

The Ethics Committee approved this study at the University Hospital of Larissa, and all subjects provided written and oral informed consent. Following consent, demographic information and data regarding past PCR-confirmed COVID-19 infection and vaccination history were recorded on questionnaire forms for all participants. Following consent, demographics, somatometric characteristics, comorbidities, medication, and data regarding past PCR-confirmed COVID-19 infection documented in the medical records, vaccination history, and previous SARS-CoV-2 antibody testing were recorded on questionnaire forms for all participants.

The SARS-CoV-2 IgG II Quant ELISA method (Architect, Abbott, IL, USA), a chemiluminescent microparticle immunoassay, was used for the qualitative and quantitative determination of IgG antibodies against the spike receptor-binding domain (RBD) of SARS-CoV-2 in serum specimens [7], with a sensitivity of 99.9% and specificity of 100% for detecting the IgG antibodies generated by prior infection or vaccination, as previously described [8].

Statistical analyses were performed with IBM SPSS Statistics for Windows, version 23.0, Armonk, NY: IBM Corp. The chi-square test and unpaired *t*-test were used to compare frequencies and parametric data between the two groups, respectively.

## 3. Results

### 3.1. Seroprevalence in Malesina

A total of 525 subjects (12% of the total population of Deskati) with a mean age of 52.5 ± 17.5 years participated in the antibody surveillance program. Females (62.7%) outnumbered males. The characteristics of the population and comparisons according to gender are shown in Table 1. Males were significantly older than females (Table 1). No difference in the other characteristics was detected. 

SARS-CoV-2 seropositivity was 77.5% (*n* = 407) in the study population. One-third of the participants had a known past infection and 41.5% of the population had been vaccinated with at least one dose of a vaccine. The seropositivity rate was significantly higher for males compared to females (Table 1). The mean antibody titer was 4005 ± 3590 AU/mL, with no difference between genders.

Number of SARS-COV-2 infections, hospitalizations and nonhospitalized cases by date in Malesina are shown in Appendix A.

Table 2 presents the comorbidities of the participants. The prevalence of hypertension, diabetes mellitus, ischemic heart disease, and heart failure was higher in men than in women. Thyroid disease was higher in females than males. Individuals with comorbidities were more frequently vaccinated than people with no comorbidities (54.6% vs. 27.0%, *p* = <0.001). There was no gender difference between the vaccinated and non-vaccinated participants. Previously infected individuals were less likely to be vaccinated than previously uninfected adults (13.8% vs. 58.2%, *p* < 0.001).

SARS-CoV-2 seropositivity was achieved by vaccine-acquired immunity only (172/407 subjects, 42.3%), naturally acquired immunity (164/407 subjects, 40.3%), or both (26/407 subjects, 6.3%), while 45/407 (11.0%) were firstly diagnosed seropositive without a known previous infection. A total of 18.9% of the participants who had received a COVID-19 vaccination were partly or fully vaccinated with the Oxford-AstraZeneca vaccine, 79.6% received at least one dose of the Pfizer-BioNTech vaccine, and 1.5% received the single-shot Johnson & Johnson vaccine.

Antibody titers were significantly higher in previously infected and vaccinated individuals than in previously infected, unvaccinated participants (14,697 ± 12,040 vs. 1917 ± 529 AU/mL, *p* < 0.001).

No age or gender difference in antibody titers was detected among vaccinated and infected populations (r = 0.168, *p* = 0.413). A mild negative correlation between age and antibody titers was detected in the only-vaccinated population (r = −0.223, *p* = 0.003). In contrast, a mild positive correlation was detected between age and antibody titers in the population who were only infected and not vaccinated (r = 0.228, *p* = 0.003).

### 3.2. Seroprevalence in Domokos

A total of 151 subjects (10% of Domokos’s population) with a mean age of 50.3 ± 13.9 years participated in the serosurveillance program. Males (55.6%) outnumbered females. The characteristics of the population and comparisons according to gender are shown in Table 3. Males were significantly older and more overweight than females (Table 2). No difference in the other characteristics was detected.

SARS-CoV-2 seropositivity was 77.5% (n = 405) in the study population. In this low-burden community, only 15.2% of the participants had a known past infection, but 66.2% of the population had been vaccinated with at least one dose of a vaccine. The was no difference in seropositivity rate between genders (Table 2). The mean antibody titers were 3284 ± 2171 AU/mL, with no difference between genders.

Table 4 presents the comorbidities of the participants. The prevalence of hypertension, diabetes mellitus, and ischemic heart disease was higher in men than women. Thyroid disease and asthma were higher in females than males. Individuals with comorbidities were more frequently vaccinated than people with no comorbidities (75.0% vs. 54.2%, *p* = 0.007). There was no gender difference between the vaccinated and non-vaccinated participants. Previously infected individuals were less likely to be vaccinated than previously uninfected adults (30.4% vs. 74.1%, *p* < 0.001).

SARS-CoV-2 seropositivity was achieved mainly by vaccine-acquired immunity (91/117 subjects, 77.8%), followed by naturally acquired immunity (14/117 subjects, 12.0%), or both (7/117 subjects, 6.0%), while 5/117 (4.2%) were firstly diagnosed seropositive having not known about the previous infection. A total of 18.1% of the participants who had received a COVID-19 vaccination were partly or fully vaccinated with the Oxford-AstraZeneca vaccine, 80.9% received at least one dose of the Pfizer-BioNTech vaccine, and 1.1% received the single-shot Johnson & Johnson vaccine.

Antibody titers had a non-significant difference between previously infected and vaccinated individuals than previously infected, unvaccinated participants (9316 ± 8321 vs. 4232 ± 3249 AU/mL, *p* = 0.132). No age difference in antibody titers was detected among only vaccinated (r = 0.012, *p* = 0.915), only infected and no vaccinated (r = −0.141, *p* = 0.629), or both vaccinated and infected populations (r = 0.494, *p* = 0.260).

## 4. Discussion

In the two years since the pandemic first emerged, there has been remarkable progress in developing an armamentarium of vaccines that are both effective and safe and are produced at scale. However, the emergence of a succession of new variants from different parts of the world has reinforced the necessity of recording country-specific immunity status along with maintaining the capability for traditional public health interventions [9]. In the present study, we evaluated the seroprevalence in two large Greek communities before the last Omicron pandemic wave—Malesina, which had been hit hard by the second Delta-driven pandemic wave, and Domokos, which had been slightly hit by the second Delta-driven pandemic wave. We found a high rate of SARS-CoV-2 seropositivity in both cities, approaching 77.5%. In the high-burden context of Malesina, seropositivity was achieved via a combination of vaccine-acquired and naturally acquired immunity, whereas in the low-burden context of Domokos, the high rates of seropositivity were achieved mainly with vaccination. Previously infected individuals were less likely to be vaccinated than previously uninfected adults. However, the antibody titers were significantly higher in previously infected, vaccinated participants than in previously infected, unvaccinated participants. Another interesting finding was that approximately 4 and 10% of the unvaccinated population were diagnosed seropositive for the first time, having not known the previous infection. Age and gender did not impact antibody titers in high- or low-burden contexts.

The question of how much SARS-CoV-2 immunized people may account for reaching herd immunity is a leading issue for expanding the debate and addressing the many raising outcries politely against vaccination [10]. Herd immunity would require more than 60 of the population to have COVID-19 immunity, either through prior infection or vaccination [11,12]. Mandatory vaccination is used to provide herd immunity but is refutable due to human rights and autonomy infringement [13]. Nevertheless, either asymptomatic or asymptomatic SARS-CoV-2 infection produces a different B-cell and T-cell memory response in vaccinated individuals. This may be due to the initial SIgA-B cell mucosal response driving a sustained IgG-B cell memory, which is the next horizon that the recent straightforward and innovative RNA-based vaccines would expect to reach. Hence, vaccines remain formidable weapons against COVID-19.

Another critical and recently emerging concept is evolving immunity secondary to vaccination. Specifically, germinal center-derived B-cells and memory B-cells have improved their avidity and affinity over time [14], indicating that as the virus evolves, it is met with a similar concept of evolving immunity. Notably, germinal center B-cell stimulation via any vaccine modality is effective against a breadth of variants compared to viral infection alone [15]. Data on booster (BNT162b2) administered to previously infected survivors indicates that this combination is efficacious at providing long-term protection [16].

As context for our findings, these studies indicate that national vaccination programs should consider the combined and variable effects of vaccination, infectious burden, and the effect of vaccination on previously infected survivors. This would allow healthcare systems to gain momentum against the pandemic by equalizing protection conferred by immunity in different settings.

Accordingly, we found that herd immunity thresholds have been reached both in low- and high-burden contexts in Greece, attributed to vaccine-acquired and naturally acquired immunity, respectively.

In areas with higher immunity because of past infections or vaccination, the number of cases of Omicron has risen quickly. However, the number of severe diseases remains low compared to areas with lower immunity. Greek authorities are urging the public to get vaccinated or get booster shots amid the Omicron wave, which has kept cases above 43,000 in the last days in order to lessen the momentum and duration of the Omicron pandemic wave.

In both contexts, 6% of the vaccinated population had a past infection. Previously infected individuals have been shown to have some humoral protection against COVID-19, although still prone to reinfection [16]. Previous studies showed that the natural infection induces robust memory T-cell responses, including long-lived cytotoxic (CD8+) T cells, with a half-life of 125 to 255 days [17].

We have previously demonstrated that vaccinating individuals post-infection substantially enhances their immune response. Furthermore, research highlights that vaccinating post-infected individuals confers strong resistance against variants of concern, such as the B.1.617.2 (Delta) variant. More specifically, it has been documented that serum spike IgG Ab levels were higher after vaccination in infected patients than after natural infection 180 days after the completion of the second vaccine dose or natural infection [18]. It is widely accepted that individuals who were both previously infected with SARS-CoV-2 and given a single dose of the vaccine gained additional protection against SARS-CoV-2 [18]. This study confirms that notion showing that the antibody titers were more than doubled in vaccinated compared with unvaccinated, previously infected participants. Similarly, in this study, we support this notion and found that antibody titers were significantly higher in previously infected and vaccinated individuals than in previously infected, unvaccinated participants. In the present study, we did not evaluate cell-mediated immunity; other studies have shown that natural infection induces a diverse polyepitope cell-mediated immune response that targets the spike protein, nucleocapsid protein, and membrane protein [19].

Another finding of this study is that approximately 4 and 10% of the unvaccinated population were firstly diagnosed seropositive for the first time, having not known any previous infection. Our team also recently reported that about two-fifths of infections determined by serology were asymptomatic, not previously known [2]. There is increasing evidence that many patients with COVID-19 are asymptomatic, but they can transmit the virus to others. Asymptomatic residents of SARS-CoV-2 are a major problem in controlling the COVID-19 pandemic in the community. The severity of asymptomatic infections was not understood and was acknowledged much later in the pandemic. There are difficulties in screening for asymptomatic infections, making it more difficult for national prevention and control of this epidemic [20]. A review based on SARS-CoV-2 infections suggested that 40% of infections were asymptomatic [21]. Once more, we highlighted the value of low-cost serosurveys targeting both symptomatic and asymptomatic populations to evaluate the immune response to SARS-CoV-2 in susceptible individuals and design evidence-based, effective and ethical response strategies to the COVID-19 pandemic. Studies in the community are the way to acquire knowledge about the prevalence of asymptomatic and mildly symptomatic cases, as well as the potency and sustainability of their acquired immune response. In addition, seroprevalence studies provide essential findings for surveillance of the evolution of the pandemic by enabling incidence estimates at the population level [22,23].

A limitation of this study was that it constituted a smaller fraction of the population in Malesina and Domokos, reaching a random sample of 10% of each population. Moreover, we cannot exclude those participants who might have characteristics associated with a willingness to participate. The population was limited to one geographic area, which translated into a lack of generalizability. A strength of this study was that individuals had been enrolled at a late phase of the Delta-driven pandemic wave, and the seroprevalence data could paint a clear picture of the subsequent immunoprotection in the targeted population.

## 5. Conclusions

Before the last Omicron pandemic wave, herd immunity was reached in low SARS-CoV-2 burden contexts in Greece, mainly due to vaccination, and in high-burden contexts either by vaccination or natural infection. Higher antibody titers are recorded in the case of vaccination in previously infected subjects compared to unvaccinated previously infected.

Effective and ethical response strategies to the COVID-19 pandemic can only be formulated once the antibody seroprevalence at the population level has been accurately determined. This study highlighted the effectiveness of epidemiological studies in continuously evaluating the baseline amount of disease occurrence and seroprevalence due to naturally or vaccine-acquired antibody production in a community.

## Figures and Tables

**Table 1 ijerph-19-06110-t001:** Characteristics of the study population in Malesina, stratified by gender (N = 525).

Variable	Total (N = 525)	Males (*n* = 150)	Females (*n* = 375)	*p*-Value
Age (years)	52.5 ± 17.5	56.1 ± 17.0	51.0 ± 17.6	0.003 ^#^
BMI (mg/kg^2^)	19.4 ± 3.1	19.8 ± 2.8	19.3 ± 3.3	0.083 ^#^
Comorbidities, *n* (%)	273 (52.0)	77 (51.3)	196 (52.3)	0.353 *
Previous infection confirmed, *n* (%)	202 (38.5)	73 (48.7)	129 (34.4)	0.017 *
Vaccinated (fully or partly), *n* (%)	218 (41.5)	65 (43.3)	153 (40.8)	0.419 *
Fully vaccinated	171 (32.6)	54 (36.0)	117 (31.2)	0.526 *
Seropositive, *n* (%)	407 (77.5)	131 (87.3)	276 (73.6)	0.003 *
Antibody titers (AU/mL)	4005 ± 3590	3675 ± 2930	4162 ± 3262	0.546 ^#^

Note: Data are expressed as mean ± SD or as frequencies (percentages). *^#^ t*-test; * chi-square.

**Table 2 ijerph-19-06110-t002:** Percentage of comorbidities (%) in Malesina’s population (N = 273).

Variable	Total (N = 273)	Males (*n* = 77)	Females (*n* = 196)	*p*-Value
Hypertension	85 (16.2)	38 (49.4)	47 (24.0)	0.001 *
Thyroid Disease	51 (19.4)	3 (3.9)	48 (24.5)	0.001 *
Diabetes mellitus	36 (13.2)	19 (24.7)	17 (8.7)	0.001 *
Hyperlipidemia	18 (6.6)	6 (7.8)	12 (6.1)	0.123 *
Ischemic Heart Disease	16 (5.9)	14 (18.2)	2 (1.0)	0.001 *
Autoimmune Disease	15 (5.5)	5 (6.5)	10 (5.1)	0.145 *
Asthma	12 (4.4)	5 (6.5)	7 (3.6)	0.055 *
Heart Failure	10 (3.7)	5 (6.5)	5 (2.6)	0.042 *
Atrial fibrillation	3 (1.1)	1 (1.3)	0 (0)	0.125 *

Note: Data are expressed as frequencies (percentages). * chi-square.

**Table 3 ijerph-19-06110-t003:** Characteristics of the study population in Domokos, stratified by gender (N = 151).

Variable	Total (N = 151)	Males (*n* = 84)	Females (*n* = 67)	*p*-Value
Age (years)	50.3 ± 13.9	52.7 ± 13.9	47.4 ± 13.4	0.018 ^#^
BMI (mg/kg^2^)	24.5 ± 10.0	28.3 ± 6.4	24.2 ± 8.2	<0.001 ^#^
Comorbidities, *n* (%)	56 (37.1)	30 (35.7)	26 (38.8)	0.412 *
Previous infection confirmed, *n* (%)	23 (15.2)	11 (13.1)	12 (17.9)	0.310 *
Vaccinated (fully or partly), *n* (%)	100 (66.2)	59 (70.2)	41(61.2)	0.160 *
Fully vaccinated	87 (57.6)	50 (59.5)	37 (55.2)	0.526 *
Seropositive, *n* (%)	117 (77.5)	67 (79.8)	50 (74.6)	0.289 *
Antibody titers (AU/mL)	3284 ± 2171	3135 ± 2003	3679 ± 2585	0.529 ^#^

Note: Data are expressed as mean ± SD or as frequencies (percentages). ^#^
*t*-test; * chi-square.

**Table 4 ijerph-19-06110-t004:** Percentage of comorbidities (%) in Domokos’ population (N = 56).

Variable	Total (N = 56)	Males (*n* = 30)	Females (*n* = 26)	*p*-Value
Hypertension	17 (30.4)	12 (40.0)	5 (19.3)	0.001 *
Thyroid Disease	8 (19.4)	1 (3.3)	7 (26.9)	0.001 *
Diabetes mellitus	5 (8.9)	4 (13.3)	1 (3.8)	0.005*
Hyperlipidemia	11 (19.6)	7 (23.3)	4 (34.6)	0.123 *
Ischemic Heart Disease	3 (5.3)	3 (10.0)	0 (0)	0.050 *
Autoimmune Disease	1 (1.8)	0 (0)	1 (3.8)	0.155 *
Asthma	6 (10,7)	1 (3.3)	4 (15.4)	0.045 *
Heart Failure	2 (3.6)	2 (6.6)	0 (0)	0.056 *
Atrial fibrillation	1 (1.8)	1 (3.3)	0 (0)	0.125

*** chi-square.

## Data Availability

The data that support the findings of this study are available on request from the corresponding author, O.S.K.

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
