# Peer review of "Lessons We Have Learned Regarding Seroprevalence in High and Low SARS-CoV-2 Contexts in Greece before the Omicron Pandemic Wave"

_ijerph, 2022, doi:10.3390/ijerph19106110_

Round 1

Reviewer 1 Report

Comment 1: In the summary section, it would be good to put background information, since they go directly to the objective of the study.

Comment 2: Why was it not considered to apply exclusion criteria to the participants?

Comment 3: Table 1 and 2; The comorbidities of the participants were not detailed.

Comment 4: Why was it not decided to determine neutralizing antibodies in the participants?

Comment 5: It would be convenient to know the percentage of neutralizing antibodies in people with and without a history of COVID-19.

Author Response

RESPONSE TO REVIEWER 1:

  1. Comment 1: In the summary section, it would be good to put background information, since they go directly to the objective of the study.

RESPONSE: Thank you for the comment. We have revised the abstract section, as suggested.

  1. Comment 2: Why was it not considered to apply exclusion criteria to the participants?

RESPONSE: Thank you for the point. Exclusion criteria were not considered as we aimed for a population-centered, epidemiological approach; i.e. an attempt to capture an accurate reflection of the population under study, rather than a specific subset.  We have added our rationale to the text.

  1. Comment 3: Table 1 and 2; The comorbidities of the participants were not detailed.

Response: Thank you for the suggestion. We have added detailed comorbidities as percentage of each areas’ population in the newly developed Table 2 and Table 4.

RESPONSE:

  1. Comment 4: Why was it not decided to determine neutralizing antibodies in the participants?

Response: Thank you for your comment. Due to logistics, we used readily available kits as provided within the setting of the Greek healthcare system. Other possible measurements were considered, but would not be feasible.

RESPONSE:

  1. Comment 5: It would be convenient to know the percentage of neutralizing antibodies in people with and without a history of COVID-19.

Response: Thank you for your comment. The reviewer’s suggestion is sound, and we would gladly consider it in a future study.

 We are grateful for the time and energy you expended on our behalf. We appreciate all of your insightful comments. We found them quite useful as we approached our revision. We very much hope that the revised manuscript now meets your expectations.

Reviewer 2 Report

Dear Authors,

the article is well written and the two small cities represent a really good choice. However, the article reported nothing more than two numbers in two essential Tables. You need to re-write a new full-article with all the clinical data in your hands and not use for other short-articles.

These are not "Lessons we have learned..." but just one "Lesson we have learned...".

At the end, I'm very sorry to disappoint 14 Authors but believe me is better to prepare a more complete article about the events in Malesina and Domokos. 

Author Response

RESPONSE TO REVIEWER 2:

Dear Authors, the article is well written and the two small cities represent a really good choice. However, the article reported nothing more than two numbers in two essential Tables. You need to re-write a new full-article with all the clinical data in your hands and not use for other short-articles.

RESPONSE: Thank you for your comment. We would argue however that in this paper, we present an epidemiological report on two rural communities and expand on how the acquired and innate immunity contribute, albeit differentially to transmission dynamics. The presentation of our data and our interpretations are supported by seroprevalence data, demographics and other variables that were reasonably able to be collected by an on-site healthcare team within the national healthcare framework of Greece. We could not consider this as an experimental report or clinical research study, but as an epidemiological study led by a point-of-care team. We hope that this clarification provides an accurate description of the strengths and limitations of our team.

These are not "Lessons we have learned..." but just one "Lesson we have learned...".

RESPONSE: Thank you for your comment. We would argue however that other than a potential rephrasing, our results explore more than one such lesson.

RESPONSE:

At the end, I'm very sorry to disappoint  Authors but believe me is better to prepare a more complete article about the events in Malesina and Domokos. 

RESPONSE: Thank you for your comment. We would argue however, that what the reviewer suggests is that we rewrite the paper so as to fit the concept of a controlled trial or clinical study with set outcomes, whereas this study reports on the findings of a point-of-care healthcare team deployed within the national framework of Greece. Such a paradigm shift is certainly not feasible, and while we are grateful for the suggestion, we cannot pursue this within a reasonable timeframe and the scope of this paper.

We are grateful for the time and energy you expended on our behalf. We appreciate all of your insightful comments.

Reviewer 3 Report

The study by Kotsiou et al reports on the seroprevalence of SARS-CoV-2 antibodies among the population of two towns in Greece. The seroprevalence was compared against demographic factors such as age, gender, and also previous infections or vaccinations. Notably, the study population represented about 10% of the town’s population. The authors show a high seroprevalence >75% in both communities however, for the town of Malesina this high seroprevalence was achieved by natural infection and vaccination equally, while for the town of Domokos it was achieved mainly by vaccination. The two towns differ as such as the first town experienced a harsh Omicron wave with many more cases compared to the latter town. This suggests better protection among the population with a higher vaccination rate.

The study is succinct and easy to follow. It is overall well-presented and the methods appear sound. However, major language editing is needed to improve the grammar and clarify different claims.

Minor comments

  1. The authors mention herd immunity but discuss numbers in that regard. Please specify what % of seropositivity you accept as herd immunity and add the appropriate references.

The authors claim that ‘herd immunity was reached before the Omicron wave’ but by definition, herd immunity would reduce subsequent transmission, which did not happen.  Please rephrase or discuss this in more detail.

  1. The authors characterize the two towns as hit hard and hit less hard but don’t specify the differences. Please add numbers on infection rates and hospitalizations for both to emphasize the different experiences of the Omicron wave for Malesina and Domokos. It might be also useful to compare that to the overall numbers for Greece.

  1. How does the seropositivity compare to the number of infections reported for these two towns before the study samples were taken? If there is data please add.

  1. The authors repeatedly refer to the 4% and 10% of unknown infections as ‘important’ or ‘major finding’. I don’t really see this as a breakthrough. Please rephrase and avoid over-emphasizing this outcome.

Suggestions for grammatical improvement:

Abstract

  1. Do not say ‘hardly hit’ or ‘hard hit’ as this implies a low impact effect. Rather use ‘hit hard’ or ‘hit badly’. The first sentence of the abstract should be rewritten, for example. In this work, we compared the SARS-C0V-2 seroprevalence in two Greek communities in June and July 2020. One community was affected worse than the other by the subsequent omicron pandemic wave that started on November 2021.
  2. Replace ‘Another interesting finding was that 4 and 10% of…’ with ‘4% and 10% of …’
  3. Rephrase the conclusion as they are a bit confusing.

Introduction

  1. End of paragraph 1. Please add the date or time frame of the implementation.
  2. Beginning of paragraph 2. ‘In our previous study we showed …’
  3. Replace ‘launched’ with ‘SARS-CoV-2 seropositivity increased to 76%’
  4. I don’t understand what ‘higher antibody titers in the case of vaccination in previously infected subjects’ refers to.

Methods

  1. Rephrase as follows: ‘a population with 1531 citizens, was not hit badly…

Results

  1. Rephrase as follows: SARS-CoV-2 seropositivity was achieved by vaccine-acquired immunity only
  2. Should read ‘'the population', 'were infected', 'not vaccinated’. titers in population who was only infected and no vaccinated (r=0.228, p=0.003).

Discussion

  1. Should read. Approximately 4% and 10% of the unvaccinated population were diagnosed seropositive for the first time,
  2. Remove ‘strongly’. In high- or low- virus burden contexts, age and gender did not strongly impact antibody
  3. Should read. This may be due to the initial SIgA‐B cell mucosal response driving a sustained IgG‐B cell memory,
  4. Should read. Another important and recently emerging concept is that of evolving immunity secondary to vaccination.
  5. Should read. We have previously demonstrated that vaccinating individuals post-infection substantially enhances their immune response.
  6. I did not understand this sentence. Also, where is the reference? Our team was also recently reported that about two-fifths of infections determined by serology are asymptomatic.
  7. I did not understand this statement. Although all individuals who participated in the present study were respondents, ….

Author Response

RESPONSE TO REVIEWER 3:

The study by Kotsiou et al reports on the seroprevalence of SARS-CoV-2 antibodies among the population of two towns in Greece. The seroprevalence was compared against demographic factors such as age, gender, and also previous infections or vaccinations. Notably, the study population represented about 10% of the town’s population. The authors show a high seroprevalence >75% in both communities however, for the town of Malesina this high seroprevalence was achieved by natural infection and vaccination equally, while for the town of Domokos it was achieved mainly by vaccination. The two towns differ as such as the first town experienced a harsh Omicron wave with many more cases compared to the latter town. This suggests better protection among the population with a higher vaccination rate. The study is succinct and easy to follow. It is overall well-presented and the methods appear sound. However, major language editing is needed to improve the grammar and clarify different claims.

 Response: Thank you for the comment. The manuscript has been revised by a professional English language editor, as suggested.

Minor comments

  1. The authors mention herd immunity but discuss numbers in that regard. Please specify what % of seropositivity you accept as herd immunity and add the appropriate references.

Response: Thank you for the remark. In the revision, we have specified what % of seropositivity we accept as herd immunity and have added the appropriate references.

  1. The authors claim that ‘herd immunity was reached before the Omicron wave’ but by definition, herd immunity would reduce subsequent transmission, which did not happen.  Please rephrase or discuss this in more detail.

Response: Thank you for the comment. This study aimed to estimate seroprevalence in two large Greek communities in June and July 2021, respectively, after the end of the Delta-driven pandemic wave that started in November 2020 up to January 2021 and before the ongoing omicron pandemic wave that started in November 2021. Accordingly, we found that herd immunity thresholds have been reached both in low and high-burden rural communities in Greece before the omicron pandemic wave, attributed to vaccine-acquired and naturally acquired immunity, respectively. There is no data regarding the subsequent transmission during the ongoing omicron pandemic wave. However, we consider it in a future study.

  1. The authors characterize the two towns as hit hard and hit less hard but don’t specify the differences. Please add numbers on infection rates and hospitalizations for both to emphasize the different experiences of the Omicron wave for Malesina and Domokos. It might be also useful to compare that to the overall numbers for Greece.

RESPONSE: This study is the first to evaluate the seropositivity in these two rural communities in Greece, and unfortunately, there is no previous data regarding the exact number of infections and hospitalizations for Domokos. This fact highlights the urgent need for epidemiological studies in Greece to enable refined exact estimates of the spread of the pandemic. Unpublished data provided by the the Civil Protection of the Municipality are presented in the following Table. Moreover, newspapers shed light on the issue of the widespread of the virus in Malesina. Moreover, in Figure 1, we compare the active cases and hospitalization of Malesina with the overall numbers for Greece. These data will be uploaded as supplementary files.

Weekly update from the Civil Protection of the Municipality of Lokroi

Date

Active SARS-COV-2 cases

Hospitalizations

Nonhopitalized cases

2021/1/4

3

1

2

2021/1/11

2

1

1

2021/1/18

5

1

4

2021/1/25

2

1

1

2021/2/1

12

1

11

2021/2/8

11

1

10

2021/2/15

17

4

13

2021/2/22

20

4

16

2021/3/1

186

11

175

2021/3/8

241

50

191

2021/3/15

164

64

100

2021/3/22

107

56

51

2021/3/29

74

42

32

2021/4/5

57

32

25

2021/4/12

45

13

32

2021/4/19

18

6

12

2021/4/26

13

2

11

2021/5/3

7

2

5

2021/5/10

14

2

12

2021/5/17

24

1

23

2021/5/24

16

3

13

2021/5/31

21

3

18

Sum

1059

301

758

Figure 1.

  1. How does the seropositivity compare to the number of infections reported for these two towns before the study samples were taken? If there is data please add.

RESPONSE: As we have previously mentioned, there is no previous estimated data regarding the seropositivity for these two communities before the study samples were measured. Again, this fact highlights the urgent need for epidemiological studies in Greece to enable refined exact estimates of the spread of the pandemic and the seropositivity of the communities.

  1. The authors repeatedly refer to the 4% and 10% of unknown infections as ‘important’ or ‘major finding’. I don’t really see this as a breakthrough. Please rephrase and avoid over-emphasizing this outcome.

RESPONSE: Thank you for the comment. In the revised manuscript, we have avoided overemphasizing this finding.

 Suggestions for grammatical improvement:

Abstract

  1. Do not say ‘hardly hit’ or ‘hard hit’ as this implies a low impact effect. Rather use ‘hit hard’ or ‘hit badly’. The first sentence of the abstract should be rewritten, for example. In this work, we compared the SARS-C0V-2 seroprevalence in two Greek communities in June and July 2020. One community was affected worse than the other by the subsequent omicron pandemic wave that started on November 2021.

Response: Thank you; We have rephrased it accordingly.

  1. Replace ‘Another interesting finding was that 4 and 10% of…’ with ‘4% and 10% of …

Response: Thank you; replaced accordingly.

  1. Rephrase the conclusion as they are a bit confusing.

Response: Thank you; rephrased accordingly.

Introduction

  1. End of paragraph 1. Please add the date or time frame of the implementation.

Response:Thank you for this point. We have added the time frame of the implementation, as suggested.

  1. Beginning of paragraph 2. ‘In our previous study we showed …’

Response: Thank you; rephrased accordingly.

  1. Replace ‘launched’ with ‘SARS-CoV-2 seropositivity increased to 76%’

Response: Thank you; rephrased accordingly.

  1. I don’t understand what ‘higher antibody titers in the case of vaccination in previously infected subjects’ refers to.

Response: Thank you for the comment. We apologize for the confusion. We clarified this point in the revised paper.

Methods

  1. Rephrase as follows: ‘a population with 1531 citizens, was not hit badly…

 Response: Thank you; rephrased accordingly.

Results

  1. Rephrase as follows: SARS-CoV-2 seropositivity was achieved by vaccine-acquired immunity only

Response: Thank you; rephrased accordingly.

  1. Should read ‘'the population', 'were infected', 'not vaccinated’. titers in population who was only infected and no vaccinated (r=0.228, p=0.003).

Response: Thank you; rephrased accordingly.

Discussion

  1. Should read. Approximately 4% and 10% of the unvaccinated population were diagnosed seropositive for the first time,
  2. Remove ‘strongly’. In high- or low- virus burden contexts, age and gender did not strongly impact antibody
  3. Should read. This may be due to the initial SIgA‐B cell mucosal response driving a sustained IgG‐B cell memory,
  4. Should read. Another important and recently emerging concept is that of evolving immunity secondary to vaccination.
  5. Should read. We have previously demonstrated that vaccinating individuals post-infection substantially enhances their immune response.
  6. I did not understand this sentence. Also, where is the reference? Our team was also recently reported that about two-fifths of infections determined by serology are asymptomatic.
  7. I did not understand this statement. Although all individuals who participated in the present study were respondents, ….

Response: Thank you for the comments. We apologize for the confusion. We have rephrased and clarified all the points in the revised paper.

We appreciate all of your insightful comments. We found them quite useful as we approached our revision. We are grateful for the time and energy you expended on our behalf. We very much hope that the revised manuscript now meets your expectations.

Round 2

Reviewer 2 Report

The article remains in the field and limited in the context of epidemiological analyzes. The current version is more adherent to the journal and some concepts are reported in a more clear way.